# Protein sequence optimization with a pairwise decomposable penalty for buried unsatisfied hydrogen bonds

Brian Coventry[1,2], David Baker[2,3,4]*

1 Molecular Engineering & Sciences Institute, University of Washington, Seattle, Washington, United States of America, 2 Institute for Protein Design, University of Washington, Seattle, Washington, United States of America, 3 Department of Biochemistry, University of Washington, Seattle, Washington, United States of America, 4 Howard Hughes Medical Institute, University of Washington, Seattle, Washington, United States of America

* dabaker@uw.edu

**Data Availability Statement:** All relevant data are within the manuscript and its Supporting Information files.

## Abstract

In aqueous solution, polar groups make hydrogen bonds with water, and hence burial of such groups in the interior of a protein is unfavorable unless the loss of hydrogen bonds with water is compensated by formation of new ones with other protein groups. For this reason, buried "unsatisfied" polar groups making no hydrogen bonds are very rare in proteins. Efficiently representing the energetic cost of unsatisfied hydrogen bonds with a pairwise-decomposable energy term during protein design is challenging since whether or not a group is satisfied depends on all of its neighbors. Here we describe a method for assigning a pairwise-decomposable energy to sidechain rotamers such that following combinatorial sidechain packing, buried unsaturated polar atoms are penalized. The penalty can be any quadratic function of the number of unsatisfied polar groups, and can be computed very rapidly. We show that inclusion of this term in Rosetta sidechain packing calculations substantially reduces the number of buried unsatisfied polar groups.

## Author summary

We present an algorithm that fits into existing protein design software that allows researchers to penalize unsatisfied polar atoms in protein structures during design. These polar atoms usually make hydrogen-bonds to other polar atoms or water molecules and the absence of such interactions leaves them unsatisfied energetically. Penalizing this condition is tricky because protein design software only looks at pairs of amino acids when considering which amino acids to choose. Current approaches to solve this problem use additive approaches where satisfaction or unsatisfaction is approximated on a continuous scale; however, in reality, satisfaction or unsatisfaction is an all-or-none condition. The simplest all-or-none method is to penalize polar atoms for simply existing and then to give a bonus any time they are satisfied. This fails when two different amino acids satisfy the same atom; the pairwise nature of the protein design software will double count the satisfaction bonus. Here we show that by anticipating the situation where two amino acids

**Funding:** The authors received no specific funding for this work.

**Competing interests:** The authors have declared that no competing interests exist.

satisfy the same polar atom, we can apply a penalty to the two amino acids in advance and assume the polar atom will be there. This scheme correctly penalizes unsatisfied polar atoms and does not fall victim to overcounting.

This is a *PLOS Computational Biology* Methods paper.

## Introduction

Polar groups on the surface of proteins in aqueous solution make favorable hydrogen bonds with water molecules. If these polar groups become buried, either upon folding or binding to another protein, these hydrogen bonds with water must be broken. The energetic penalty of losing h-bonds with water can be offset if a buried polar atom makes a hydrogen bond to another protein atom. We say that this second polar atom "satisfies" the first polar atom. If when buried, the first atom does not make a hydrogen bond, we call it a "buried unsatisfied".

Modeling the loss in favorable interactions of buried unsatisfied polar atoms is straightforward with explicit solvent models since upon burial, interactions with explicit water molecules are lost. For protein design and other applications where large scale sampling is required and chemical composition (amino acid identity) is changing, implicit solvation models have considerable advantage over explicit models in computational efficiency. The most computationally efficient implicit solvent models are pairwise additive, but identifying and penalizing buried unsatisfied polar atoms is challenging using such models as burial is a collective property. Instead, most current methods use non-pair additive approaches, often involving solvent accessible surface area calculation. The BuriedUnsatsfiedPolarCalculator in Rosetta for instance first calculates which atoms are inaccessible to solvent, and then determines whether or not they are making a hydrogen bond. These methods work well on a fixed protein, but they are not amenable to the implicit-solvation pairwise-decomposable sidechain packer of Rosetta[1] or other rotamer-based packing algorithms.

There have been several attempts to capture the energetic cost of buried unsatisfied polar atom penalty in a pairwise manner. The LK solvation model gives all polar atoms a penalty when another atom enters its implicit sphere of solvation[2]. The LK-Ball solvation model takes the LK solvation model and restricts it to positions most critical for hydrogen bonding [3]. A downside to these pairwise methods is that they are intrinsically additive. Instead of a switching behavior where an atom becomes completely buried and can no longer hydrogen bond with water, the burial is gradual and depends on the local density of nearby atoms. These approaches do not specifically penalize buried unsatisfied polar atoms, but instead attempt to model this effect indirectly through balancing the energies of desolvation and hydrogen bonding.

## Materials and methods

We describe a method for explicitly penalizing buried unsatisfied polar atoms during sidechain packing called the 3-Body Oversaturation Penalty (3BOP). We take advantage of the fact that for rotamer-based side chain packing calculations, while the calculated energies must be pairwise-decomposable, the calculations leading to these energies need not be pairwise-decomposable. The method is applicable to fixed-backbone packing trajectories and requires that all rotamers are available before the Monte Carlo trajectory begins.

## Burial region calculation

To assign the buried region in a sequence-independent way so that it can be determined before amino acid sequence design, the protein is first mutated to poly Leucine with Chi1 = 240 and Chi2 = 120. Next, the EDTSurf method[4,5] is used to voxelize cartesian space at 0.5Å resolution, determine the molecular surface with a 2.3Å radius sphere, and label each voxel with its depth below the molecular surface. The burial region is then defined as all voxel XÅ below the molecular surface where X is between 3.5–5.5Å depending on user preference. Fig 1B shows an example of a 4.5Å burial region on Ubiquitin[6].

## Penalty calculation

After the burial region calculation, all buried polar atoms in all rotamers are identified. One-body and two-body atom pseudoenergies can then be assigned with the following simple algorithm:

```
for B in all_buried_polar_atoms
  Constants β, σ, ω
  Accumulate 1-body energy β to B
  for Q in atoms_that_hbond_to_B:
    Accumulate 2-body energy σ to edge B<->Q
  for Q1, Q2 in pairs of (atoms_that_hbond_to_B):
    Accumulate 2-body energy ω to edge Q1<->Q2
```

Constants $\beta$, $\sigma$, and $\omega$, representing the atom burial penalty, atom-atom satisfaction bonus, and atom-atom oversaturation penalty may be selected on a per-buried-polar-atom basis according to Eq 1. Fig 1A gives a pictorial illustration of this algorithm, which is O ($n^3$) on the local number of polar rotamers that h-bond at nearby sequence positions; the algorithm is approximating a 3-body interaction.

## Oversaturation rotamer correction

A problem occurs with this simple algorithm when multiple B from different rotamers at the same sequence position hbond to the same Q1<->Q2 pair. With each additional B, the oversaturation penalty between Q1 and Q2 rises. This oversaturation penalty is an error because these B cannot exist at the same time. The solution is to limit the oversaturation penalty to the maximum value one rotamer can generate at each position.

Corrected 3BOP Algorithm:

```
for n in [1 ... N_res]
  Let M = map (key = Rotamer<->Rotamer, value = list (rotamers_at_n))
  for R in rotamers_at_n
    for B in buried_polar_atoms_of_R
      Constants β, σ, ω
      Accumulate 1-body energy β to B
      for Q in atoms_that_hbond_to_B:
        Accumulate 2-body energy σ to B<->Q
      for Q1,Q2 in non-redundant-paired (atoms_that_hbond_to_B):
        M [r_Q1<->r_Q2] [R] + = ω
  for r_Q1<->r_Q2 in M.keys()
    Accumulate 2-body energy max (M [r_Q1<->r_Q2]) to r_Q1<->r_Q2
```

where r_Q refers to the rotamer containing Q. The memory footprint of M can be greatly reduced if instead of storing a list, one only stores the running max value after iterating over each rotamer R.

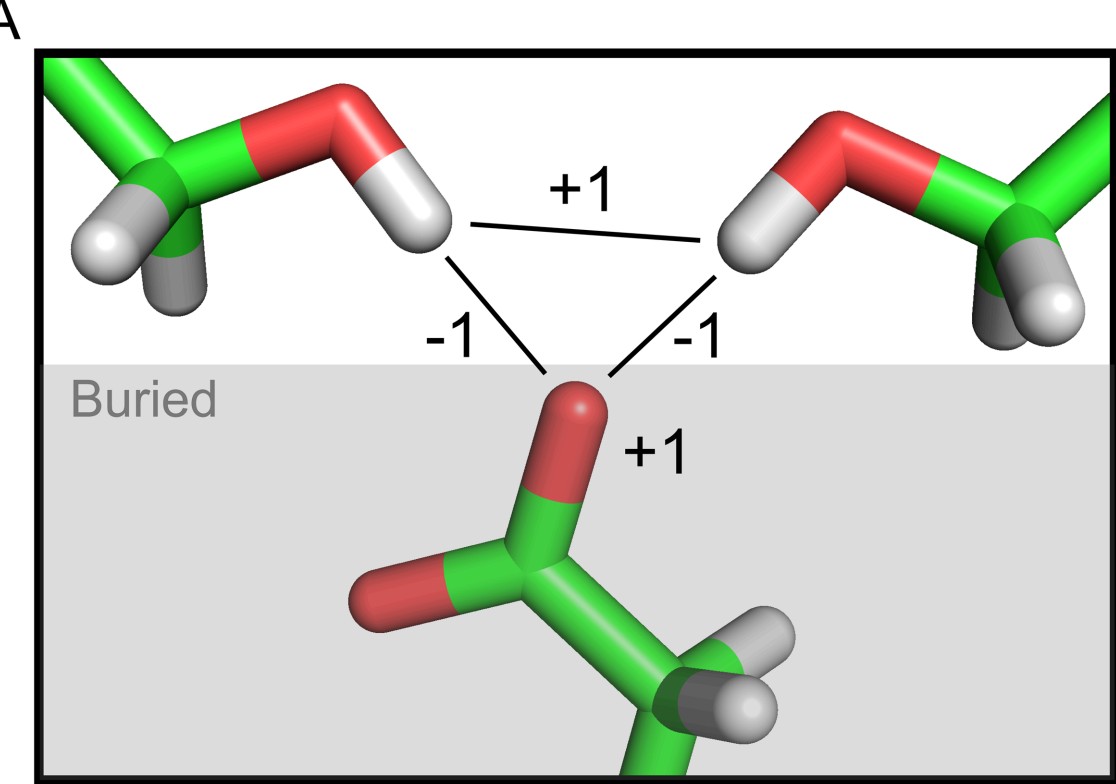

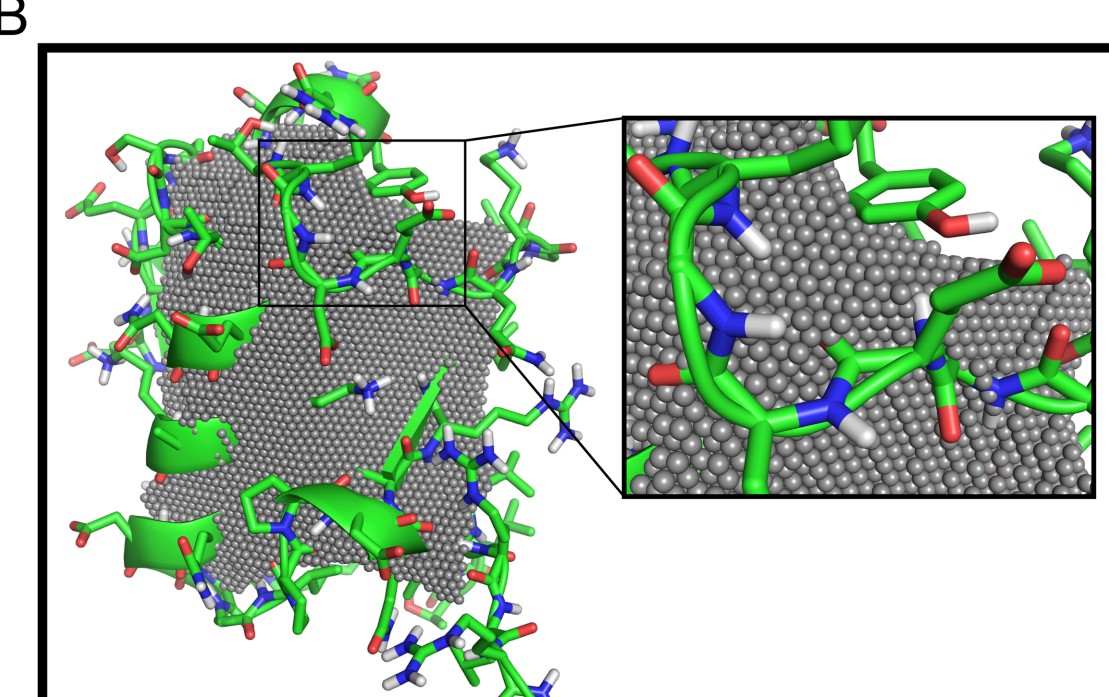

**Fig 1. Overview of method** (A) Pictorial representation of the penalty rules. A buried GLU oxygen atom is satisfied by two serine hydroxyls. An oversaturation penalty is applied to the two hydroxyls. In this example: $\beta = 1$, $\sigma = -1$, $\omega = 1$. (B) Burial region calculation applied to Ubiquitin (PDB: 1UBQ)[6]. Grey spheres indicate the buried region at 4.5Å depth from the poly-LEU molecular surface with a 2.3Å probe size and were generated using PyRosetta[7]. Images generated with PyMOL[8].

## Results

### Buried unsatisfied polar quadratic penalty

The 3BOP algorithm described in the Materials and Methods generates a penalty P of the form

$$P = \beta + \sigma \cdot H + \omega \cdot \frac{H \cdot (H - 1)}{2} \qquad (1)$$

where H is the number of h-bonds, $\beta$ is the penalty for burying a polar atom, $\sigma$ is the bonus for satisfying a buried polar atom, and $\omega$ is the penalty for oversaturating a buried polar atom. Because of the quadratic term, this formulation can better describe the "all or none" aspect of buried unsatisfied atoms than linear models such as the LK solvation model. The coefficients $\beta$, $\sigma$, and $\omega$ can be modified to give any quadratic relationship between the number of h-bonds and the penalty. Additionally, they can be modified on a per-atom basis to give different penalty profiles to different atoms types. As Table 1 shows, parameters can be chosen in general to favor any single number or pair of consecutive numbers of h-bonds.

Since the atomic depth calculation is performed on a poly-Leucine backbone, there is no dependence upon the sequence or sidechain conformations to determine burial; hence atomic burial can be pre-computed once before packing begins. However, polar atoms just below the surface will not be considered buried. Consider for instance a backbone carbonyl oxygen that is outside of the burial region, but that is covered by a Phenylalanine ring. Such an oxygen would be buried by explicit solvent or SASA-based measure, but would not be buried by this algorithm.

### Fewer buried unsatisfied polar atoms

Four hundred *de novo* one-sided interfaces to barnase[9] were designed to test the effectiveness of 3BOP in generating designs that make hydrogen bonds to buried polar atoms. Designed mini-proteins[10] docked to the polar interface of barnase had fewer buried unsatisfied polar atoms when 3BOP was added to the ref2015[11] energy function (Fig 2A). With the recommended setting of 5, the 50th percentile structure coming from 3BOP + ref2015 had 5 buried unsatisfied polar atoms while the 50th percentile structure coming from ref2015 alone had 7. While a reduction from 7 to 5 may seem small, it is important to note that many of the docks had impossible-to-satisfy polar atoms (e.g. edge-strand to hydrophobic surface). Some docks fully satisfied all polar atoms on the target; 3BOP + ref2015 generated 3 such docks while ref2015 alone generated none. As a further test, one hundred native proteins were redesigned to only allow polar residues as described in S1 Text. S1A Fig shows that redesigning the proteins with 3BOP + ref2015 resulted in a median of 2 buried unsatisfied polar atoms while ref2015 alone resulted in a median of 5 (the 3BOP designs also had fewer polar atoms; S1C Fig).

**Table 1. Example penalty schemes for different atom types.**

| Atom Type | Target #H-Bonds | Coefficients | Resulting Penalty for Buried Polar Atom | | | | | |
|---|---|---|---|---|---|---|---|---|
| NH1 | 1 | $\beta = 1, \sigma = -1, \omega = 2$ | 1 | 0 | 1 | 4 | 9 | 16 |
| Carbonyl O | 1 or 2 | $\beta = 1, \sigma = -1, \omega = 1$ | 1 | 0 | 0 | 1 | 3 | 6 |
| NH2 | 2 | $\beta = 4, \sigma = -3, \omega = 2$ | 4 | 1 | 0 | 1 | 4 | 9 |
| Carboxylate O | 2 or 3 | $\beta = 3, \sigma = -2, \omega = 1$ | 3 | 1 | 0 | 0 | 1 | 3 |
| NH3 | 3 | $\beta = 9, \sigma = -5, \omega = 2$ | 9 | 4 | 1 | 0 | 1 | 4 |
| | | # h-bonds | 0 | 1 | 2 | 3 | 4 | 5 |

Examples of coefficients that lead to a penalty of 1 for buried polar atoms that are off-by-1 from their ideal number of h-bonds. The target # Hbonds listed here are for example only and are not necessarily ideal.

### 3BOP is better than simply upweighting h-bonds

Figs 2A and S1A show that incorporation of 3BOP results in fewer buried unsatisfied polar atoms than simply increasing the hydrogen bond strength. However, a limitation of our approximation is that the oversaturation penalties do not depend on the presence of the buried polar atom rotamer. For instance, in Fig 1, if the Glutamate rotamer was not present, the penalty between Serine rotamers would still be applied. To investigate the extent to which this happens, Outer Membrane Phospholipase A of E. Coli, a natural protein with extensive buried polar networks was repacked with an implementation of 3BOP in Rosetta. As S2A Fig shows, the less strict the threshold for h-bonds, the more the extraneous penalties. As the number of rotamers increase, either by adding extra rotamers, or enabling design, the number of extraneous penalties further increases. This problem may be reduced by increasing the stringency of the h-bond-quality threshold or limiting the number of polar rotamers.

While the extraneous oversaturation penalties may pose a problem for structure prediction, their effect on protein design is not as severe. For designed proteins, there may be several solutions that satisfy a backbone and if one of them is eliminated by an error, there are still several other equally valid solutions, and an almost infinite number of backbones may be attempted. Even if all solutions for a backbone are eliminated another backbone will likely provide solutions. Overall, for design, sins of omission (moving forward with designs containing buried unsats) are more serious than sins of commission (incorrectly eliminating a reasonable design).

## Discussion

An advantage of an explicit penalty for buried unsatisfied polar atoms is that protein designers can set the penalty to values appropriate to their application. Previous approaches have sought to more accurately model the energetics of protein electrostatics, but the final functional forms are intertwined with the rest of the force-field in a way that could not be arbitrarily modified. Explicit control allows designers to directly penalize buried unsatisfied polar atoms. Designers also have control over the level of hydrogen bond satisfaction in their designs. For example, the number of h-bonds that the NH2 group of glutamine (1, 2, or 1 or 2) must make to be considered satisfied can be specified by suitable parameter choices (Table 1). While this paper and the implementation in Rosetta do not consider explicit bound water molecules in crystal structures or from other sources, incorporating these is straightforward. As long as the location of the water molecules is known at pre-compute time, they may simply be modeled as polar atoms that can make hydrogen bonds. The 3BOP algorithm is computationally efficient (S3 Fig), and in its current form without explicit water atoms, is now widely used in our research group, and we expect that it will quite broadly help address long standing issues with buried unsaturated polar atoms in de novo protein design.

## Supporting information

**S1 Fig. Effect of penalty on protein redesign.** 100 native proteins were redesigned using the energy function and protocol described at the left of the panels, allowing only polar sidechains. Each method/row uses the same parameters as Fig 2A except Lysine-NZ used $\beta = 15$ and $\sigma = -10$ for the "5 x" variants and $\beta = 30$ and $\sigma = -20$ for the "10 x" variants. A) Number of buried unsatisfied polar atoms for each protein. In order from the left, vertical divisions indicate the number of proteins that have 0, 1, 2, or more unsatisfied polar atoms as indicated in the last row. B) Number of h-bonds to buried polar atoms. In order from left, each division represents the number of proteins that had from X to (X+4) h-bonds to buried polar atoms with each division to the right representing from (X+5) to (X+9). C) Number of buried polar atoms. In

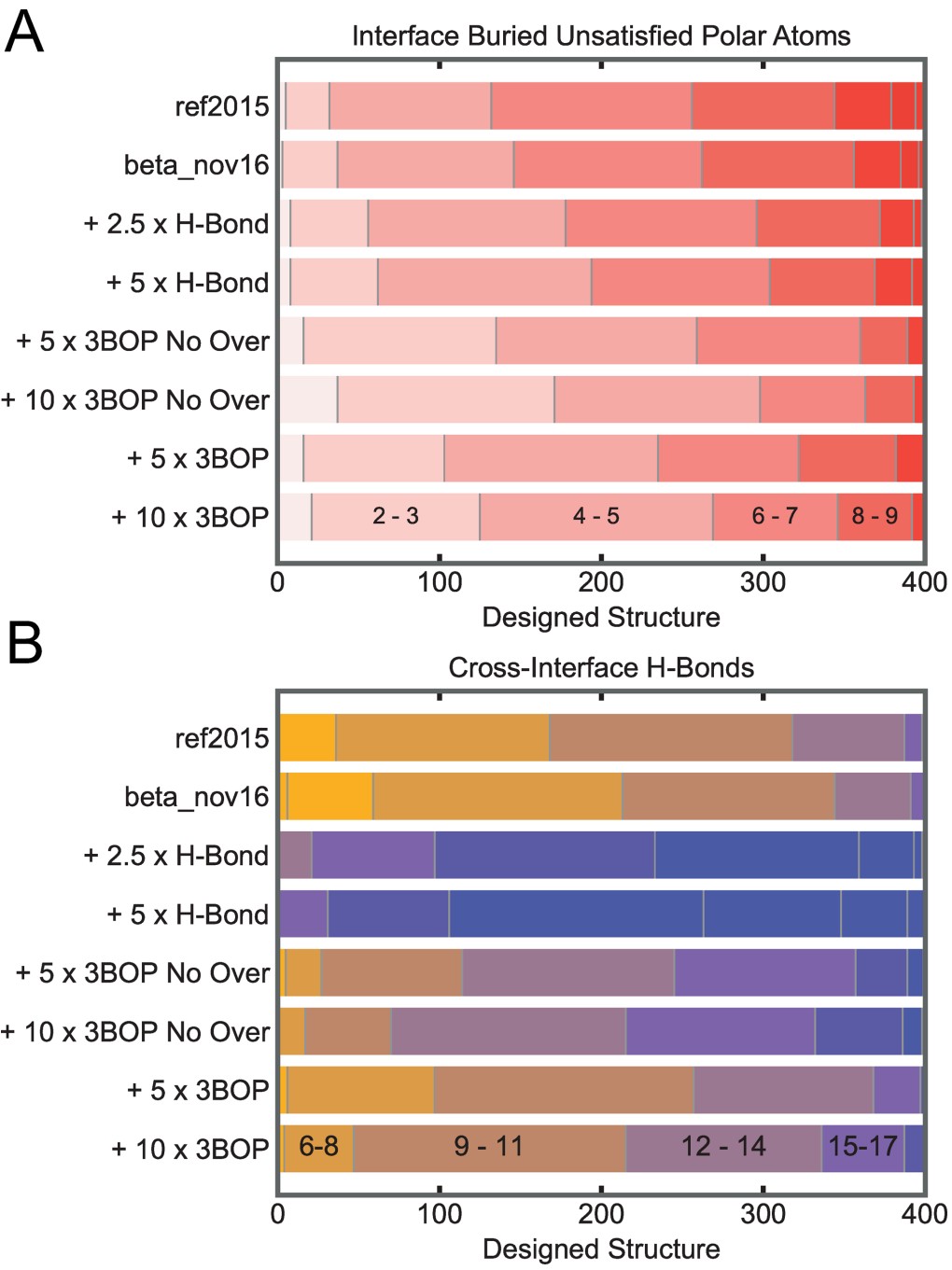

**Fig 2. Effect of 3BOP on *de novo* interface design.** 40 mini-proteins of mixed alpha/beta and all alpha topology were docked against barnase (PDB: 1BRS) using PatchDock[12,13] and the top 10 docks for each selected. Residues within 10Å of the interface were allowed to repack on barnase and to design with all 20 amino acids on the mini-protein. A ramped-repulsive pack and minimize scheme was used to arrive at the final amino acid sequence[14] using the score function described at the left of the panels. All methods that start with "+" use ref2015. For "+ 5 x 3BOP No Over", the parameters were β = 5, σ = -5, and ω = 0 and for "+ 5 x 3BOP", the parameters were β = 5, σ = -5, and ω = 5 for all polar atoms. The "+ 10 x" variants used 10 instead of 5 for the respective parameters. As the maximum energy for an h-bond in Rosetta is approximately -2 kcal/mol, the "+ 2.5 x H-Bond" and "+ 5 x H-Bond" increased the maximum h-bond energy to -7 and -12 kcal/mol respectively. A) Number of buried unsatisfied polar atoms at the interface. In order from the left, vertical divisions indicate the number of proteins that have 0–1, 1–2, 2–3, or more unsatisfied polar atoms as indicated in the last row. B) Number of cross-interface h-bonds. In order from left, each division represents the number of proteins that had from X to (X+2) cross-interface h-bonds which each division to the right representing (X+3) to (X +5). See S1 Text for more information and S1 Scripts for scripts to reproduce the data in S1 Data.

order from left, each division represents the number of proteins that had from X to (X+9) buried polar atoms with each division to the right representing from (X+10) to (X+19). For more information, see S1 Text.
(TIF)

**S2 Fig. Extraneous oversaturation and performance** A) The Outer Membrane Phospholipase A (PDB: 1ILZ) [15] was either repacked with standard rotamers (purple plus) or extra rotamers (pink cross) or redesigned with all amino acids using standard rotamers (green down arrow) or extra rotamers (yellow up arrow). An expansive buried h-bond network exists in the structure. The percentage of native rotamers in this h-bond network that experience extraneous oversaturation penalties to other native rotamers is plotted vs the energy threshold for a h-bond to be considered. In short, the extraneous oversaturation penalties were determined by performing the 3BOP algorithm and looking for penalties between native rotamers that were not present before the design/repack rotamers were considered (see S1 Text). The black line shows the percentage of h-bonds in the h-bond network that pass the energy threshold. B) Ninety-seven native proteins had their h-bond network residues redesigned using only polar amino acids. Amino acid recovery error of ref2015 (grey), ref2015 + 5 x 3BOP No Over (light blue), and ref2015 + 5 x 3BOP (dark blue) plotted. Parameters used for 3BOP tests identical to Fig 2 except Lysine-NZ used $\beta = 15$ and $\sigma = -10$. The h-bond threshold was set to -0.75. See S1 Text for details.
(TIF)

**S3 Fig. Performance of Rosetta-implementation of 3BOP.** Each stacked bar graph represents the CPU time spent performing a packing or design calculation on 1ILZ using the pre-computed interaction graph setting. The red top bar represents time spent applying the penalty rules to rotamers, green bar represents time spent calculating h-bonds between rotamers before the 3BOP algorithm, the orange bar is time spent calculating atomic depth, and the blue bottom bar is runtime of the background packing or design calculation. With better data management, the green bar could be avoided as h-bonds are double calculated here (with the other calculation occuring inside blue). *While 3BOP adds a large runtime penalty here, only 2% of this runtime is spent calculating the actual 3-body interactions. 98% of the runtime is spent later in dictionary lookups during rotamer-pair energy assignment.
(TIF)

**S1 Text. Data collection for figures.**
(PDF)

**S1 Scripts. Scripts to produce results.**
(TAR)

**S1 Data. Results data.**
(TAR)

## Acknowledgments

We thank Vikram K. Mulligan and Scott Boyken for helpful thoughts and conversations about buried unsaturated polar atoms and the idea of creating an energetic penalty for these.

## Author Contributions

**Data curation:** Brian Coventry.

**Investigation:** Brian Coventry.

**Methodology:** Brian Coventry.

**Supervision:** David Baker.

**Validation:** Brian Coventry.

**Visualization:** Brian Coventry.

**Writing – original draft:** Brian Coventry.

**Writing – review & editing:** David Baker.

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
