## [Decision Letter · Decision Letter 0]

26 Aug 2020

Dear Mr. Coventry,

Thank you very much for submitting your manuscript "Protein sequence optimization with a pairwise decomposable penalty for buried unsatisfied hydrogen bonds" for consideration at PLOS Computational Biology.

As with all papers reviewed by the journal, your manuscript was reviewed by members of the editorial board and by several independent reviewers. In light of the reviews (below this email), we would like to invite the resubmission of a significantly-revised version that takes into account the reviewers' comments.

We cannot make any decision about publication until we have seen the revised manuscript and your response to the reviewers' comments. Your revised manuscript is also likely to be sent to reviewers for further evaluation.

Sincerely,

Dina Schneidman

Software Editor

PLOS Computational Biology

Reviewer's Responses to Questions

**Comments to the Authors:**

Reviewer #1: The authors present an approximation algorithm to the problem of penalizing the burial of polar atoms when those atoms don't form hydrogen bonds. This problem cannot be readily expressed with terms of a pairwise decomposable energy function. If a polar atom is considered buried, then it can be assigned a penalty, and then hydrogen bonds to that polar atom can be assigned a bonus to compensate for that penalty. However, if multiple hydrogen bonds form to a single atom, they would all get that bonus and instead of simply compensating for the burial of the polar atom, they keep accumulating their bonuses. In the context of protein design, this would produce sequences that form an unrealistic number of hydrogen bonds.

The authors approach this problem by putting oversaturation penalties between pairs of atoms that hydrogen bond to the same third atom. The calculation of this penalty depends on three things at once (three atoms / three rotamers) and is therefore not pairwise decomposable; however, this calculation can occur before the beginning of the sequence optimization step. The oversaturation penalty between the pairs of atoms that hydrogen bond to the same third atom is included in the pair interaction energy of those two atoms and will be included whether or not the third atom is present.

This is a clever approach to a vexing problem, and the authors present some compelling results, but there are a few things that leave the reader unsatisfied.

1. The authors describe their oversaturation penalty as "the algorithm" throughout the text. It needs a name. "Algorithm" is too general a term -- and isn't strictly speaking even the right term for the approach the authors present. "Oversaturation penalty" perhaps? "Three-body oversaturation penalty" (3BOP)? This is a minor point and wouldn't be the first in the list except that it makes it a little harder to write this review. I will refer to the authors approach as "3BOP" for the remainder of this review.

2. The authors don't really tell us what parameters to use for 3BOP and leave out some important details about the experiments that they performed. For example, figures 2A and 2B both include two rows "+ 5*Algorithm" and "+10*Algorithm" without much in the way of an indication what 5 and 10 mean or what values for the three parameters that define 3BOP (b, s, & v) they are using (or whether they are using different values for each of the atom types?). Furthermore, it is not even clear which of the two (5 vs 10) the authors are endorsing as the best. (The authors do send the readers to the supplemental materials in the caption to this figure, but something as pertinent to the understanding of a figure should not required going to the supplement.)

The "complete-protein, polar only" design test presented in Figure 2 is an interesting one. At first glance, it seems silly, since no protein designer would ever design a protein that lacked hydrophobic residues, but at second glance, it seems like a clever way to prevent an algorithm that wants to avoid burying unsatisfied polar atoms from avoiding polar atoms altogether. It is fascinating and confusing that the median number of buried polar atoms per design could decrease from 5 to 2 while the number of hydrogen bonds only increased by two and certainly looks better than the version of the score function that simply increases the strength of hydrogen bonds. However, on third glance, the 3BOP still has an "out" that the polar-only setup doesn't perfectly avoid: it is possible for 3BOP to create fewer buried unsatisfied polar atoms by simply designing fewer polar atoms: it could choose poly serine and do better than the default score function, ref2015, which might try to use larger residues that coincidentally contain more polar atoms. It seems important to report the number of buried polar atoms that the different score functions create.

A better test would be 1-sided interface design and interrogate the number of buried unsatisfied polar atoms on the side held fixed. This test case has the further advantage of being much closer to the purpose where 3BOP is likely to be employed.

3. Figure 3B shows that only a tiny fraction of the total design time is consumed on 3BOP; however, 3BOP relies on the pre-calculation of all pairs of rotamer energies ahead of design, and the standard design algorithm in rosetta is to compute pair energies on the fly. The authors should mention whether 3BOP can be adapted to be used in an on-the-fly energy calculation scheme and, if not, how much of a performance penalty is incurred switching between on-the-fly and pre-calculation.

4. Figure 3A shows a concerning presence of oversaturation penalties between pairs of atoms that are simultaneously hydrogen bonding to the same absent/phantom atom. It seems like there is likely some price that comes with the use of 3BOP, but there's little that the authors present in terms of investigations into what that price would be. Does it impact the probability of seeing certain amino acid pairs at certain CB distances, perhaps?

5. The choice of letters and formalism in the pseudo-code is confusing. Typically a set would be labeled with a capital letter, (e.g. S) and an element of that set would be labeled with a lower case letter (e.g. s). In the pseudo code, capital letter B is used to denote an element of the set of buried polar atoms, and the lower case letter b is used to denote the one-body penalty for a buried polar atom. Perhaps use greek letters for constants: beta, sigma and omega? Furthermore, the "Q1-Q2" or "B-Q" notation looks like subtraction instead of denoting the pair interaction energy between the atoms.

**Have all data underlying the figures and results presented in the manuscript been provided?**

Reviewer #1: Yes

PLOS authors have the option to publish the peer review history of their article (what does this mean?). If published, this will include your full peer review and any attached files.

Reviewer #1: No
---

## [Decision Letter · Decision Letter 1]

12 Feb 2021

Dear Mr. Coventry,

We are pleased to inform you that your manuscript 'Protein sequence optimization with a pairwise decomposable penalty for buried unsatisfied hydrogen bonds' has been provisionally accepted for publication in PLOS Computational Biology.

Best regards,

Dina Schneidman

Software Editor

PLOS Computational Biology

Reviewer's Responses to Questions

**Comments to the Authors:**

Reviewer #1: The authors have satisfied all of my requests with their review.

**Have all data underlying the figures and results presented in the manuscript been provided?**

Reviewer #1: None

PLOS authors have the option to publish the peer review history of their article (what does this mean?). If published, this will include your full peer review and any attached files.

Reviewer #1: No

---

## [Editor Report · Acceptance letter]

2 Mar 2021

PCOMPBIOL-D-20-01008R1 

Protein sequence optimization with a pairwise decomposable penalty for buried unsatisfied hydrogen bonds

Dear Dr Coventry,

I am pleased to inform you that your manuscript has been formally accepted for publication in PLOS Computational Biology. Your manuscript is now with our production department and you will be notified of the publication date in due course.

With kind regards,

Alice Ellingham
